# Anti-Inflammatory Pathways Mediating tDCS’s Effects on Neuropathic Pain

**DOI:** 10.3390/biology14070892

**Published:** 2025-07-20

**Authors:** Haipeng Zhang, Xinyan Zheng, Binn Zhang

**Affiliations:** 1School of Exercise and Health, Shanghai University of Sport, 200 Hengren Road, Yangpu, Shanghai 200438, China; 2321518004@sus.edu.cn; 2School of Psychology, Shanghai University of Sport, 399 Changhai Road, Yangpu, Shanghai 200438, China

**Keywords:** neuropathic pain, tDCS, inflammatory mediators, neuroplasticity, glial, cytokines, cortex, spinal cord, dorsal ganglion root, peripheral nerve

## Abstract

Neuropathic pain (NP) is a common clinical health problem caused by an injury or illness that affects the somatosensory system. Its therapeutic efficacy is unsatisfactory because it does not seem to be effectively treated by most analgesics. The developmental factors and mechanisms of NP are complex and diverse. Transcranial direct current stimulation (tDCS), one of the most commonly used non-invasive brain stimulation methods, is widely used in the treatment of NP. Numerous studies have shown that tDCS is reliable in the treatment of NP. However, current studies suggest that tDCS may reduce NP symptoms by regulating inflammatory mediators, but the regulatory mechanism has not been clearly and uniformly explained. This review summarizes the mechanism by which tDCS improves NP by regulating inflammatory mediators. tDCS has shown significant potential in the treatment of NP by targeting the multi-level neuromodulation of neuroinflammation, synaptic plasticity, and ion channel dynamics. By linking molecular mechanisms with systems-level neural regulation, tDCS is expected to develop into a precision medicine cornerstone for refractory NP.

## 1. Introduction

Neuropathic pain (NP) is a debilitating condition resulting from lesions or diseases in the somatosensory system, presenting a significant clinical challenge due to its resistance to conventional analgesic treatments [1]. It is characterized by allodynia and hyperalgesia, and affects approximately 7–8% of the general population, accounting for 20–25% of chronic pain cases. This imposes substantial burdens on individuals and healthcare systems alike [2]. The complex pathogenesis of NP involves peripheral and central sensitization, as well as neuroimmune interactions, highlighting the urgent need for innovative therapeutic strategies [3].

Among the emerging non-invasive therapies, transcranial direct current stimulation (tDCS), a form of non-invasive brain stimulation (NIBS), has gained attention for its potential to modulate the maladaptive neural plasticity associated with chronic pain [4]. tDCS involves delivering a low-intensity current (typically 0.5–2 mA) via scalp electrodes to induce subthreshold shifts in neuronal membrane potentials, thereby altering cortical excitability and synaptic efficacy [5,6]. Initial studies by Nitsche and Paulus [7] demonstrated that anodal tDCS over the primary motor cortex (M1) enhances corticospinal excitability, a mechanism leveraged for pain management. Subsequent research expanded to other regions, such as the dorsolateral prefrontal cortex (DLPFC), reflecting the role of cognitive–affective circuits in pain processing [8].

Emerging evidence suggests that neuroinflammatory pathways play a crucial role in mediating the effects of tDCS on neuropathic pain. Key inflammatory mediators, including tumor necrosis factor-α (TNF-α), interleukin-1β (IL-1β), and interleukin-10 (IL-10), which are regulated by astrocytes and microglia, exhibit significant dysregulation in NP and correlate strongly with pain intensity [9]. Preclinical studies consistently show that tDCS administration reduces spinal IL-1β and hippocampal TNF-α levels while increasing IL-10 expression [10,11], suggesting a multi-level anti-inflammatory action. However, the variability in clinical outcomes, potentially resulting from differences in stimulation protocols or patient selection criteria, underscores the challenges in translating these findings into clinical practice [12].

This review aims to synthesize the current evidence on the mechanisms through which tDCS alleviates neuropathic pain, focusing specifically on its modulation of inflammatory mediators. We will evaluate the region-specific effects of tDCS on neuroimmune signaling pathways in the brain, spinal cord, and dorsal root ganglia. Additionally, we will explore the role of glial–neuron interactions in tDCS-mediated analgesia and discuss the critical translational challenges in optimizing tDCS protocols for various subtypes of neuropathic pain.

## 2. Therapeutic Effects of tDCS on Neuropathic Pain Management

tDCS is a non-invasive neuromodulation technique that influences the brain’s resting membrane potential by applying a steady, low-intensity current (0.5–2 mA), leading to excitatory alterations in neural activity [13]. This approach has shown promise in the management of chronic pain, particularly NP, due to its potential to modulate abnormal neural plasticity. According to a 2017 evidence-based guideline, anodal tDCS targeting the M1, or the hemisphere opposite to the side of pain, is recommended as a potentially beneficial intervention for managing lower limb pain resulting from spinal cord lesions (Level C) [14]. Another guideline issued by 12 further supports the use of anodal M1 tDCS, assigning it a B-level recommendation due to its probable efficacy in alleviating NP. Despite these recommendations, the therapeutic efficacy of tDCS in NP treatment remains mixed, with varying outcomes across different types of NP.

Relevant studies were ultimately identified through searches of academic databases, including PubMed, Web of Science, Cochrane Library, Embase, and EBSCO, performed in March 2025, with no restriction on the publication year and language. The search strategy used a mix of Mesh (or publication type) and free text terms from the following two groups: 1. “tDCS [Mesh]”, “Anodal Stimulation Transcranial Direct Current Stimulation”, “Anodal Stimulation tDCS*”, “Cathodal Stimulation Transcranial Direct Current Stimulation”, “Cathodal Stimulation tDCS*”, “Transcranial Alternating Current Stimulation”, “Transcranial Random Noise Stimulation”, “Repetitive Transcranial Electrical Stimulation”, and “Transcranial Electrical Stimulation*”; and 2. “Neuralgias [Mesh]”, “Neurodynia*”, “Neuropathic Pain*”, “Nerve Pain*”, “Paroxysmal Nerve Pain*”, “Atypical Neuralgia*” “Iliohypogastric Nerve Neuralgia*”, “Ilioinguinal Neuralgia *”, “Perineal Neuralgia*”, “Stump Neuralgia*”, “Supraorbital Neuralgia*”, and “Vidian Neuralgia*”. To ensure robustness in evaluating human studies, we have set the inclusion and exclusion criteria as follows. The inclusion criteria include (1) patients diagnosed with neuropathic pain (no restrictions on sex and age); (2) the interventions were tDCS interventions; (3) the comparators were sham stimulation; (4) the outcome measure was changes in pain intensity; and (5) the study design was an RCT. The exclusion criteria include (1) duplicate items and (2) articles that were reviews, clinical registrations, animal studies, conference proceedings, pilot studies, case reports, editorial letters, or the full text was not available. Figure 1 presents the flow chart of the search for bibliographic references for this review.

We ultimately included a total of 11 randomized controlled trials. Table 1 presents the therapeutic effects of tDCS with different parameters and stimulation sites on NP patients. Several studies have demonstrated promising results for tDCS in patients with NP related to spinal cord injury (SCI), highlighting its excellent therapeutic effects [15,16]. However, not all studies report positive outcomes. For instance, a randomized controlled trial by Lewis [17] found no significant effect of 1 mA anodal M1 tDCS on patients with upper limb NP. Conversely, Bolognini [18] demonstrated that anodic M1 tDCS (2 mA for 15 min) could effectively reduce short-term phantom limb pain. The study indicated that further investigation into the clinical effectiveness of tDCS for specific NP subtypes is warranted. One critical factor influencing the therapeutic outcome of tDCS is the choice of the stimulation target region. M1 has been the focal point of most studies due to its key role in the pain matrix and its involvement in pain processing [19,20]. The evidence supports the efficacy of M1-tDCS in treating various NP conditions, including SCI-related NP [15,16], post-stroke pain [21], multiple sclerosis-related pain [22,23], phantom pain [18], trigeminal neuralgia [24], and Parkinson’s disease-related pain [25].

Beyond M1 stimulation, emerging research has investigated the therapeutic potential of targeting other brain regions with tDCS for NP management. tDCS targeting the dorsomedial prefrontal cortex (DLPFC), which is connected to brain structures such as the periaqueductal gray, thalamus, and amygdala, has also been explored for NP management [8]. A randomized controlled trial of patients with painful diabetic peripheral neuropathy (PDPN) revealed significantly greater analgesic effects following DLPFC stimulation compared to sham interventions, though the pain reduction was significantly less pronounced than that achieved with M1 stimulation [26]. However, in a study that conducted tDCS stimulation in the DLPFC area for treating multiple sclerosis-related pain, we observed a significant improvement in pain intensity compared to the sham stimulation group [27]. These findings highlight the need for further exploration of the other brain regions in the context of tDCS for pain management.

In addition to tDCS alone, several studies have explored combination therapies that integrate tDCS with behavioral treatments. This approach is based on the premise that behavioral therapy may enhance the neural circuits activated by tDCS, thereby potentiating the plasticity effects of neuromodulation [28]. For example, Kumru et al. [29] reported a 50% reduction in pain intensity in 13 participants who received daily tDCS combined with visual deception over two weeks. Similarly, a crossover trial by Boggio [30] found that combining tDCS with transcutaneous electrical nerve stimulation (TENS) resulted in more significant pain reduction compared to either treatment alone. Moreover, Soler [31] demonstrated that combining tDCS with visual illusion therapy effectively reduced pain severity in SCI-related NP, with the combined treatment yielding longer-lasting benefits compared to the individual therapies. These studies provide strong evidence for the potential of combining tDCS with behavioral interventions as a promising approach for chronic pain management. However, further research is necessary to optimize these combination strategies and assess their efficacy across diverse NP subtypes.

The differential efficacy of tDCS across various subtypes of NP is likely attributable to fundamental distinctions in their underlying pathophysiological mechanisms. Phantom limb pain is predominantly associated with central neural reorganization, characterized by pronounced cortical plasticity abnormalities, rendering the stimulation of M1 more responsive in eliciting analgesic effects [32]. In contrast, peripheral NP involves distinct neural circuits, wherein modulation via M1-targeted tDCS may exert a comparatively limited therapeutic impact [33]. Variability in the stimulation intensity, as well as inter-individual differences in cortical excitability, have also been implicated as potential confounding factors influencing treatment outcomes. Validation of these observations requires further investigation through tDCS interventions tailored to specific NP subtypes, with an additional focus on optimizing stimulation parameters such as the intensity, frequency, and duration.

The variability in analgesic efficacy observed across different stimulation targets is hypothesized to result from the functional specialization of these brain regions in pain processing. The M1 area is integrally connected with pain perception networks, including the thalamus, brainstem, and spinal cord, suggesting a more direct influence on nociceptive signal modulation and, consequently, a more pronounced analgesic response. In comparison, the DLPFC is primarily implicated in the affective and cognitive dimensions of pain, indicating a greater potential for alleviating anxiety and discomfort, but a relatively limited capacity for direct pain suppression [34]. There is a pressing need for large-scale, multicenter, randomized controlled trials to systematically evaluate the therapeutic efficacy of various cortical targets in the treatment of neuropathic pain.

While combination therapies integrating tDCS with behavioral interventions show promising initial results [15,31,32], several fundamental limitations currently constrain the interpretation of these findings. First, the majority of studies fail to implement adequate sham controls for both the neuromodulation and behavioral components, making it difficult to isolate specific treatment effects from placebo responses. Second, the substantial variability in reported effect sizes (ranging from 30 to 70% pain reduction) suggests potential confounding by unmeasured variables such as treatment adherence, pain chronicity, or individual differences in neuroanatomy. Most critically, the putative neural mechanisms underlying these synergistic effects remain largely hypothetical. For example, this mechanistic uncertainty is well illustrated by Kumru et al.’s visual illusion study [31], where the observed analgesic effects likely resulted from attentional modulation through the frontoparietal circuitry rather than the intended sensorimotor network targeting. Such a fundamental discrepancy in mechanism engagement has profound implications for developing precisely targeted combination therapies.

**Table 1 biology-14-00892-t001:** Therapeutic effects of tDCS treatment for NP based on human experiments.

Studies	Design	Participants		Intervention			Outcomes
		N (M/F)	Type of Injury	tDCS Group	Stimulated Area	Application Parameters	
Ngernyam et al. [15]	Randomized controlled trial	20 (15/5)	Spinal cord injury	2 mA anodal tDCS	M1	A single session of 20 min	Pain intensity (NRS): ↓
Fregni et al. [16]	Randomized controlled trial	17 (14/3)	Spinal cord injury	2 mA anodal tDCS	M1	20 min for 5 consecutive days	Pain intensity (VAS): ↓
Thibaut et al. [22]	Randomized controlled trial	33	Spinal cord injury	2 mA anodal tDCS	M1	Phase II: 10 sessions a day during weekday for 2 weeks	Pain intensity (VAS): ↓
Bolognini et al. [18]	Randomized controlled trial	8 (3/5)	Phantom limb pain	2 mA anodal tDCS	M1	A single session for 15 min	Pain intensity (VAS): ↓
Bae et al. [21]	Randomized controlled trial	14 (7/7)	Post-stroke pain	2 mA anodal tDCS	M1	3 sessions per week for 20 min during 3 consecutive weeks	Pain intensity (VAS): ↓
Kim et al. [25]	Randomized controlled trial	20	PDPN	2 mA anodal tDCS	DLPFC	20 min for 5 consecutive days	Pain intensity (VAS): ↓
Young et al. [23]	Randomized controlled trial	30	Multiple sclerosis	2 mA anodal tDCS	M1	20 min once a day during 5 consecutive days	Pain intensity (VAS): ↓
Ayache et al. [27]	Randomized controlled trial	16	Multiple sclerosis	2 mA anodal tDCS	DLPFC	20 min for 3 consecutive daily session	Pain intensity (BPI and VAS): ↓
Hagenacker et al. [24]	Randomized controlled trial	10	Trigeminal neuralgia	1 mA anodal tDCS	M1	20 min for 2 weeks	Pain intensity (VRS): ↓
Lewis et al. [17]	Randomized controlled trial	30 (21/9)	Upper limb injury	1 mA anodal tDCS	M1	20 min for 5 consecutive days	No significant improvement in pain intensity (BPI).
González-Zamorano et al. [25]	Randomized controlled trial	22 (10/12)	Parkinson’s disease-related pain	2 mA anodal tDCS	M1	10 sessions of 20 min for 10 consecutive days	Pain intensity (KPPS): ↓

Abbreviation: M, male. F, female. tDCS, transcranial direct current stimulation. M1, primary motor cortex. DLPFC, dorsomedial prefrontal cortex. PLP, phantom limb pain. VAS, visual analogue scale. BPI, Brief Pain Inventory. VRS, Verbal Rating Scale. KPPS, King’s Parkinson’s Pain Scale. ↓, significant decrease.

## 3. Inflammatory Mechanisms of tDCS for NP

Maladaptive neuroplasticity plays a critical role in NP pathogenesis and maintenance. However, substantial evidence indicates that NP mechanisms extend beyond neuronal dysregulation to encompass complex interactions among neurons, neuroimmune cells, and immune-like glia. Research indicates that activated glial cells release a variety of pain-sensitive substances (such as inflammatory cytokines), which exert effects on the glial cells themselves, neurons within the myelin sheath, and primary afferent neurons, thereby enhancing pain transmission [35,36,37]. Neural injury triggers immune responses observed across multiple anatomical sites: injured nerves, dorsal root ganglia (DRG), the spinal cord, and supraspinal regions within pain pathways. Overall, the persistence of post-neural injury pain is also fundamentally influenced by the equilibrium between pro- and anti-inflammatory immune mechanisms. Shifting this balance toward anti-inflammatory processes could establish novel therapeutic windows for halting chronic neuropathic pain progression. Glial cells may offer a novel strategy for the treatment of inflammatory pain [38,39,40,41]. However, a novel theoretical construct termed “dual pro-plasticity” was recently proposed in a 2024 study, where the authors postulated that inflammation triggered by neural injury induces neuroplastic alterations [42]. These pathological changes subsequently exacerbate glial activation and inflammatory cascades, establishing a self-perpetuating cycle of pathological neuroplasticity and neuroinflammation. The “dual pro-plasticity” theory is a relatively comprehensive theoretical model that provides a general overview of the etiological mechanism of NP.

tDCS is characterized as a multimodal therapeutic approach, with its efficacy in NP management potentially achieved through central neuromodulation mechanisms. The therapeutic mechanisms of tDCS in NP management are characterized by multilevel neuromodulation spanning the central and peripheral nervous systems. At the cerebral level, tDCS-mediated neuroplasticity modulation is evidenced in the hippocampus and motor cortex, where an enhanced astrocytic glutamate uptake capacity facilitates glutamine synthesis to reduce excitotoxicity. Concurrently, N-methyl-D-aspartate receptor (NMDAR)-dependent synaptic plasticity is promoted through the activation of the calcium/ inositol trisphosphate (IP3) signaling pathway in astrocytes, with observations of the suppression of peripheral nociceptive afferent inputs and enhanced excitability of the cerebral cortex. Spinal cord investigations reveal that tDCS inhibits the release of pro-inflammatory factors such as IL-1 and TNFα by M1-type microglia to reduce inflammation and relieve pain, and the transformed M2-type microglia release IL-10 to enhance the inhibitory effect of pro-inflammatory factors, achieving the effects of anti-inflammation and pain relief. Within DRGs, the therapeutic effect of tDCS is attributed to the downstream inhibitory effect on sodium channels mediated by the upregulation of IL-10 at the spinal cord level. Peripheral pain modulation is predominantly mediated through spinal-level downstream regulation, where IL-10 upregulation suppresses pro-inflammatory factors to attenuate nociceptive signaling, thereby reducing immune cell recruitment and activation at injury sites and within the central nervous system. Despite these advances, critical knowledge gaps persist regarding the precise signaling cascades, cross-hierarchical regulatory networks, and clinical translation efficacy, necessitating further mechanistic exploration to optimize tDCS-based NP intervention strategies.

### 3.1. Brain Inflammatory Response

The brain plays a central role in the generation, maintenance, and regulation of NP. Following central nervous system (CNS) injury, regions such as the cerebral cortex, medulla oblongata, and thalamus experience chronic inflammation, accompanied by corresponding physiological changes in electrical signals [43,44,45,46]. Astrocytes, located between neurons in the nervous system, play a crucial role in supporting and separating neural cells. They actively provide nutrients to the brain and spinal cord, maintain homeostasis, and regulate synaptic transmission [47,48,49]. However, the dysregulation of astrocytic calcium signaling in the anterior cingulate cortex (ACC) has been related to excessive glutamatergic synaptic transmission, exacerbating mechanical hypersensitivity in animal models [50]. Astrocytic excitability, driven by fluctuations in intracellular calcium (Ca^2+^) levels, enables these cells to respond to synaptic activity through the activation of glutamate receptors [51]. Increased Ca^2+^ levels trigger astrocytes to function as the primary synaptic glutamate scavengers through specific transporters, namely, excitatory amino acid transporters type 1 and 2 (EAAT1/GLAST and EAAT2/GLT-1). They also provide neurons with glutamine, which is utilized for the synthesis of glutamate required for neurotransmission, thereby contributing to the glutamate–glutamine cycle [52]. Ca^2+^-dependent exocytosis of glutamatergic gliotransmitters in turn, signals adjacent neurons [53]. In a study using astrocyte-specific mGluR5 conditional knockout (astro-mGluR5 cKO) mice, transient re-expression of mGluR5 in cortical astrocytes after partial sciatic nerve ligation (PSNL) induced synaptic remodeling, which contributed to mechanical allodynia [54]. Specifically, mGluR5 activation in S1 cortical astrocytes elevated Ca^2+^ signaling, promoting the expression of synaptogenic molecules such as thrombospondin-1 (TSP1), glypican-4, and hevin [55]. These molecules drive the formation of hyperexcitable synapses, which in turn induce persistent neuronal hyperactivity, ultimately contributing to refractory neuropathic pain [55].

The hippocampus is a key brain region involved in pain regulation, and integrates nociceptive, affective, and cognitive components [56,57]. The hippocampus has been increasingly associated with the therapeutic effects of tDCS [58,59]. Zin et al. demonstrated that a single session of tDCS alleviated NP symptoms by inducing neurochemical changes in the hippocampus, including the modulation of the inflammatory response [60]. Increased glutamine synthetase (GS) activity was observed in tDCS-treated animals, substantiating astrocytic involvement in electrostimulation-induced neuroprotection. The glutamine level has been considered to be involved in mediating the effect of tDCS on alleviating the pain symptoms of NP [61]. Specifically, tDCS enhanced the astrocytic glutamate uptake capacity and facilitated subsequent glutamine synthesis, leading to reduced excitotoxicity. This action possibly elevates pain thresholds and mitigated NP symptomatology in rats, and it helps to inhibit the excessive accumulation of glutamate, which in turn prevents the excessive excitation of neurons [60].

It is hypothesized that tDCS may additionally facilitate long-term potentiation (LTP) through NMDAR pathway modulation via glial activation, enhancing motor cortical neuronal excitability. In humans, tDCS over the motor cortex increases cortical excitability in an NMDAR-dependent manner [62]. Similarly, animal studies have shown that tDCS enhances synaptic responses in motor cortical slices, requiring NMDAR activation and BDNF signaling [63]. Studies have shown that astrocytes are critical mediators of NMDAR-dependent plasticity, primarily through the secretion of synaptogenic signaling molecules via gliotransmission [64,65,66,67]. Although the precise mechanisms of gliotransmission remain debated, the evidence suggests a link to elevated intracellular Ca^2+^ levels [68,69]. Calcium imaging studies further identified astrocytes as the primary cellular substrate for tDCS-induced cortical Ca^2+^ surges, with the Ca^2+^/inositol trisphosphate (IP3) signaling pathway playing a critical role in these elevations [70]. Given the positive correlation between astrocytic Ca^2+^ levels and extracellular levels of D-serine (an NMDAR co-agonist), it is hypothesized that tDCS enhances NMDAR-dependent synaptic plasticity via astrocyte-specific Ca^2+^/IP3 pathway activation by serine agonists. This mechanism not only supports gliotransmission but also holds therapeutic potential for attenuating NP-associated allodynia and hyperalgesia.

Moreover, Takeda et al. reported that tDCS of the somatosensory cortex modulated astrocytic Ca^2+^ activity and performed corrective remodeling of the synapses of S1 neurons, possibly contributing to the resolution of mechanical allodynia and normalization of tactile sensitivity [71]. Specifically, low-intensity tDCS enhanced the calcium ion activity of early-stage S1 brain astrocytes to activate S1 cortical astrocytes while suppressing nociceptive peripheral afferent inputs, correcting maladaptive synaptic remodeling in S1 circuits [71]. Such neural remodeling is postulated to preferentially eliminate maladaptive nociceptive circuits, thereby permanently resolving mechanical allodynia and restoring normative tactile processing. This represents a conceptual paradigm shift in chronic pain therapeutics: rather than merely mitigating consequences of neuronal hyperexcitability and synaptic transmission in hyperactive circuits, the therapeutic focus is redirected toward astrocyte-mediated cortical circuit plasticity—potentially reversing mechanisms underlying the transition from acute to chronic pain states. The schematic diagram illustrating the mechanisms of NP etiology and the therapeutic pathway of tDCS in relation to brain inflammation is presented in Figure 2.

### 3.2. Spinal Cord Inflammatory Response

Spinal neuroinflammation plays a pivotal role in nociceptive signaling, characterized by the release of pro-inflammatory mediators and activation of glial cells—predominantly microglia and astrocytes—which exert distinct yet complementary roles in amplifying pain signals following peripheral nerve injury [72,73,74]. Microglia, the macrophage-like cells of the central nervous system, play a crucial role in maintaining homeostasis under physiological conditions. The sensitization of microglia represents a key element in the pathogenesis of inflammatory pain [75,76]. Tissue damage activates microglia, leading to the secretion of various pro-inflammatory cytokines, such as IL-1β, interleukin-6 (IL-6), and TNF-α [77,78,79]. The release of these pro-inflammatory cytokines contributes to the persistence of behavioral hypersensitivity induced by injury or surrounding inflammation and plays a critical role in chronic inflammatory pain [80,81]. Specifically, according to a recent study [10], the CCI model increased the IL-1 levels in spinal cord for approximately 30 days following surgery. A negative correlation—in which a higher thermal threshold is connected with lower levels of IL-1—was observed between spinal cord IL-1 levels and thermal hyperalgesia at 24 h. This finding demonstrates the critical function that IL-1 plays in mediating the connection between the central nervous system’s immune response and the control of harmful behavior. After the spinal cord inflammatory response occurs, microglia, upon stimulation, will become either the M1 type or the M2 type [82]. The M1 type exhibits pro-inflammatory and cytotoxic characteristics. It releases inflammatory factors such as IL-1β, TNF-a, etc. The M2 type, on the other hand, exhibits anti-inflammatory functions and can secrete anti-inflammatory cytokines such as IL-10. In animal studies, we found that microglia in the brains of rats with spinal cord injury were activated, and the proportion of M1 type cells was significantly increased. This change mainly occurred in the ventral posterior lateral (VPL), ventral striatum (VTA), and periaqueductal gray matter (PAG) regions of the brain [83]. Overall, it can be inferred that the inflammatory response in spinal cord may be related to the increase in the levels of pro-inflammatory factors released by M1 microglia cells.

Emerging evidence suggests that tDCS may attenuate spinal neuroinflammation as a therapeutic strategy for NP after spinal cord injury. The data showed that tDCS decreased the spinal cord levels of IL-1 that had been elevated by the CCI model and successfully alleviated the nociceptive behavior until at least seven days after the treatment’s conclusion [84]. Furthermore, studies have revealed that TNF-alpha production in the spinal cord can be decreased by tDCS. The inflammatory reaction in the spinal cord may be suppressed and NP may be relieved by reducing the production of these cytokines [10]. However, a previous study has shown that tDCS altered microglial activation, thereby suppressing spinal inflammatory responses [85]. We speculate that the observed suppression of elevated pro-inflammatory factor levels appears to be associated with phenotypic transformation of microglia. A study investigated the effects of tDCS on neuropathic pain induced by spinal cord injuries, and significant improvements were noted in the thermal stimulation threshold and motor function of rats following tDCS treatment compared to the control group. Notably, it was observed that tDCS reduced the proportion of the M1 phenotype of microglia within the VPL, VTA, and PAG regions, while increasing the proportion of the M2 phenotype [83]. tDCS effectively reduced the concentrations of pro-inflammatory cytokines in the cortex, thalamus, midbrain, and medulla, while increasing the concentration of the anti-inflammatory cytokine IL-10 in the thalamus. Previous studies have shown that M1 phenotype microglial cells can transition to the M2 phenotype [86]. It is speculated that under tDCS stimulation, the polarization of microglia toward the M1 phenotype is suppressed in the aforementioned brain regions, while polarization toward the M2 phenotype is promoted. By inhibiting the release of pro-inflammatory cytokines such as IL-1 and TNF-α from M1 phenotype microglia, inflammation is reduced and pain is alleviated, whereas the transformed M2 phenotype microglia enhance the inhibitory effects on pro-inflammatory responses through the release of IL-10, thus achieving anti-inflammatory effects and pain relief. Overall, the methods through which tDCS controls the inflammatory response in the spinal cord during the treatment of NP are intricate and varied. However, current evidence supporting the efficacy of tDCS in treating chronic NP associated with spinal cord injury remains limited [87]. Robust clinical and mechanistic investigations are required to validate tDCS-mediated anti-inflammatory effects on NP symptom management. The schematic diagram illustrating the mechanisms of NP etiology and the therapeutic pathway of tDCS in relation to spinal cord inflammation is presented in Figure 3.

### 3.3. Dorsal Root Ganglion Inflammatory Response

Pain and hypersensitivity are transmitted via the initial afferent fibers, reaching the DRG for impulse transmission. DRGs constitute a critical anatomical locus in the pathogenesis of neuropathic pain. The hypersensitivity of primary sensory neurons within DRGs has been implicated as a pivotal contributor to neuropathic pain development [88,89]. Recent studies reveal that post-neural injury alterations in ion channel expression induce neuronal hyperexcitability and sustained activation within DRGs, further perpetuating chronic pain perception [90,91]. TNF-α is released by a variety of cells, including inflammatory, immune, glial, and neuronal cells [92,93]. The pro-inflammatory cytokine TNF-α has been demonstrated to upregulate voltage-gated sodium channels in DRG neurons, driving spontaneous action potential generation and hyperexcitability, which may underlie ectopic mechanical sensitivity and neuropathic pain progression [94]. Moreover, elevated BDNF (brain derived neurotrophic factor) levels in the DRG have also been associated with enhanced pain transmission, hyperalgesia, and central sensitization [95]. After neural injury, increased BDNF expression in TrkC and TrkB receptors has been confirmed [96]. Meanwhile, activated and recruited macrophages within DRGs have been identified as critical contributors to BDNF production [97,98,99,100]. These macrophages exert key roles in initiating and maintaining mechanical hypersensitivity in NP via BDNF-TrkB signaling [98,99]. BDNF promotes the transmission of pain signals within neurons by establishing synaptic connections through the Trkb signaling pathway. Collectively, injury-induced BDNF release and expression facilitate central sensitization and NP pathogenesis, highlighting its therapeutic relevance.

The regenerative capacity of DRGs, as recently highlighted in NP research, provides critical insights into innovative therapeutic strategies. IL-10, a pivotal anti-inflammatory cytokine, not only suppresses pro-inflammatory cytokine production but also inhibits their downstream effects [101]. Recent research indicated that tDCS elevates IL-10 levels in the hippocampus and sciatic nerve of NP rats post-intervention. The phenomenon is closely associated with pain behaviors and the manifestation of peripheral sensitization symptoms [11]. Although the study did not measure IL-10 levels in the DRG, the elevated IL-10 levels observed in the spinal cord and sciatic nerve suggest that IL-10 exerts its downstream effects by inhibiting action potential generation at the synapses within the DRG. We speculate that this effect is likely related to the suppression of neuronal excitability. Shen et al. [102] demonstrated in animal models that IL-10 may attenuate neuropathic pain via the downregulation of sodium channels in DRG neurons. The underlying mechanism may involve the release of a substantial amount of anti-inflammatory factors by the DRG to counteract the upregulation of sodium channels induced by TNF-α following peripheral injury.

In spinal circuits, prior studies have confirmed that glial cell-derived BDNF drives neuronal hyperexcitability by sensitizing primary afferents, a mechanism implicated in NP pathogenesis [103]. Research reported that elevated BDNF levels in DRG neurons correlate with increased spinal dorsal horn BDNF levels in inflammatory pain models [95,104]. Notably, tDCS has been shown to reverse the spinal BDNF elevation, alleviating mechanical allodynia and hyperalgesia in rodent models [105]. It is hypothesized that this spinal cord-level effect likely extends downward to the DRG, where it exerts an inhibitory effect on neuronal excitability through BDNF. A proposed mechanism involves the tDCS-mediated modulation of BDNF/TrkB inhibitory signaling cascades extending from the spinal cord to DRGs, though the precise neurobiological pathways remain undefined. A meta-analysis of the role of BDNF in tDCS suggests that tDCS may induce the expression and release of BDNF in the spinal dorsal horn, which binds to its high-affinity receptor, TrkB [106]. This interaction initiates a signaling cascade related to long-term potentiation (LTP), facilitating the reorganization of new synaptic connections through TrkB signaling and thereby promoting the long-term enhancement of cortical neuronal excitability [106]. Further investigations are required to elucidate tDCS-driven BDNF regulation in DRG neuroinflammation and validate its therapeutic potential for NP management. The schematic diagram illustrating the mechanisms of NP etiology and the therapeutic pathway of tDCS in relation to dorsal root ganglion inflammation is presented in Figure 4.

### 3.4. Peripheral Nerve Inflammatory Reaction

Pain and hypersensitivity originate from the impulses of primary afferent fibers (Aβ and C fibers). A neuroinflammatory reaction is the key to the formation of peripheral sensitization. Schwann cells play a critical role in axonal regeneration following peripheral nerve injury. After peripheral nerve damage, Schwann cells can be activated into immune-competent cells, producing inflammatory cytokines such as tumor necrosis factor-α, interleukin-6, and interleukin-1β [107,108,109,110]. It has been observed that the inflammatory response following peripheral nerve injury facilitates the transmission of pain signals to the central nervous system through interactions between neurons and glial cells. For instance, following trigeminal nerve injury, trigeminal ganglion neurons generate a series of action potentials, which are transmitted to the caudal subnucleus of the trigeminal nerve (Vc) and the upper cervical spinal cord (C1/C2). Subsequently, various molecules are produced within the trigeminal ganglion neurons. These molecules are released from the neuronal cell bodies and transported to both the central and peripheral terminals of the neurons. These alterations lead to changes in the neuronal plasticity of the trigeminal ganglion neurons, rendering them hypersensitive. This heightened neuronal activity further accelerates, leading to the sensitization of Vc and C1/C2 neurons. Satellite glial cells contribute to the enhancement of neuronal activity in the trigeminal nerve, Vc, and C1/C2 neurons [111].

The mechanisms by which tDCS can relieve NP by regulating the inflammatory reaction of peripheral nerves are not yet fully understood, but several proposed mechanisms are available. In a study investigating tDCS treatment in NP rats, it was found that tDCS may alter neuronal activity through inflammatory mediators, modifying the biochemical characteristics of the sciatic nerve and ultimately leading to pain relief [11]. A study found that the sciatic nerve’s reduced IL-10 levels remarkably affect sensory function and motor recovery [112]. The results of an ELISA demonstrated significant reductions in IL-10 levels in the sciatic nerve and DRG of chronic constriction injury (CCI) and partial sciatic nerve ligation (PSL) rats at postoperative days 3 and 8 compared to naïve controls, concomitant with marked pain-related behaviors [113]. The observed reduction in IL-10 levels may indirectly exacerbate nociceptive behaviors, potentially modulated by peripheral and central neural activity. Sacerdote et al. [9] validated temporal alterations in IL-10 expression within the peripheral nervous system of neuropathic pain models. Following sciatic nerve injury, the inflammatory mediators that are released peripherally activate glial cells, which release a large number of pro-inflammatory cytokines, triggering an inflammatory response and initiating a series of pro-inflammatory cascades [72,73]. At this point, the balance of inflammation is disrupted. IL-10 protein levels are decreased despite upregulated mRNA expression. This contrasting change can be interpreted as the rapid utilization of proteins to counteract the pro-inflammatory cascade induced by injury while simultaneously activating synthesis mechanisms to support the sustained anti-inflammatory effects of IL-10 [113,114]. Notably, elevated levels of IL-10 were detected in the spinal cord and sciatic nerve of the tDCS-treated mice, suggesting that tDCS may regulate IL-10 levels as an inflammatory mediator to alleviate neuropathic pain [11]. However, the exact role of IL-10 in tDCS-mediated neuropathic pain remains unclear. Nonetheless, substantial evidence suggests that IL-10 primarily inhibits nociceptive pain by suppressing pro-inflammatory cytokines, thereby reducing the recruitment and further activation of immune cells at the site of injury and within the central nervous system. Another study had shown that tDCS can modulate the activity of descending inhibitory pathways that start in the brain and extend to the spinal cord [115]. These pathways can inhibit the activity of pain-transmitting neurons in the spinal cord, thereby alleviating pain and inflammation in the peripheral nerve. This may provide supporting evidence for the increased IL-10 levels observed in the above study. Consequently, IL-10 warrants further investigation as a therapeutic target for mitigating neuroinflammation in neuropathic pain (NP) pathogenesis.

Moreover, the current evidence indicates that tDCS can alleviate pain and improve nerve function by altering other biochemical profiles in the sciatic nerve to reduce the production of inflammatory markers [11]. Specifically, tDCS induced neuroplastic changes in the brain that can exert downstream effects on peripheral nerves. For example, tDCS can increase the activity of brain-derived neurotrophic factor (BDNF), which can promote the growth and survival of peripheral nerves and reduce inflammation [105]. However, there are no more detailed research reported on the regulatory mechanism of tDCS in this field. Overall, tDCS can be a promising technique for the treatment of NP by regulating the inflammatory reaction of peripheral nerves. Considerable research is required to completely comprehend the underlying mechanisms and improve therapeutic regimens. The schematic diagram illustrating the mechanisms of NP etiology and the therapeutic pathway of tDCS in relation to peripheral nerve inflammation is presented in Figure 4.

## 4. Translational Limitations and Future Directions

The translation of tDCS’s effects from animal models to clinical applications for neuropathic pain faces several significant challenges. Animal studies predominantly utilize standardized nerve injury models that induce mechanical allodynia, while human neuropathic pain encompasses diverse etiologies with complex psychosocial dimensions that are difficult to replicate in rodents [1,3]. This fundamental difference in pain phenotypes limits the generalizability of preclinical findings to the heterogeneous clinical population. Species-specific variations in neuroimmune responses present another critical barrier. While rodent studies demonstrate consistent patterns of glial activation and cytokine modulation following tDCS, human neuroinflammatory pathways may respond differently due to the evolutionary divergence in immune system regulation [73,116]. The blood–brain barrier’s differential permeability across species further complicates the direct translation of anti-inflammatory effects observed in animal models. Methodological disparities between preclinical and clinical studies contribute to translational challenges. Animal research typically employs prolonged, low-intensity stimulation protocols that differ substantially from clinical tDCS parameters in both the dose and application [10,14]. The lack of standardized protocols for the current density adjustment across species makes it difficult to determine equivalent effective doses between rodents and humans. Outcome measurement discrepancies represent an additional limitation. Preclinical studies rely primarily on evoked pain behaviors, whereas clinical trials must incorporate subjective pain reports and functional assessments that reflect the multidimensional nature of the human pain experience [2]. This measurement gap underscores the need for bridging biomarkers that can objectively track treatment effects across species, such as neuroimaging markers of cortical excitability or serum cytokine profiles [117].

Future research directions should prioritize the development of more clinically relevant animal models that incorporate cognitive and affective pain components. Advanced techniques such as patient-derived xenograft models or humanized mouse systems could help bridge the species gap [85]. Multimodal intervention strategies combining tDCS with pharmacological or behavioral therapies may enhance the translational potential by addressing the complexity of human neuropathic pain [28]. The establishment of standardized reporting guidelines for tDCS studies across species would facilitate more meaningful comparisons between preclinical and clinical findings. Collaborative consortia could develop unified protocols for stimulation parameters, outcome measures, and data analysis methods [12]. Longitudinal studies tracking both molecular and clinical outcomes will be essential to validate the durability of tDCS’s effects observed in animal models. Addressing these translational challenges requires a concerted effort to align preclinical and clinical research paradigms. By developing more sophisticated animal models, implementing standardized protocols, and incorporating translational biomarkers, future studies can better evaluate the therapeutic potential of tDCS for neuropathic pain management. The integration of computational modeling with experimental data may further enhance our ability to predict and optimize tDCS’s effects across species [118].

## 5. Conclusions

tDCS demonstrates multimodal therapeutic potential for NP by concurrently targeting neuroinflammation and synaptic plasticity, with bidirectional crosstalk between these two regulatory mechanisms. The intervention appears to simultaneously suppress pro-inflammatory cytokines (IL-1β and TNF-α) while enhancing anti-inflammatory mediators (IL-10), creating a favorable microenvironment for neural reorganization. This immunomodulation interacts bidirectionally with tDCS-induced neuroplastic changes, including NMDA receptor-dependent synaptic remodeling mediated through astrocytic calcium/IP3 signaling pathways. The concurrent regulation of ion channel homeostasis in the dorsal root ganglia suggests a comprehensive “top-down” mechanism spanning the central and peripheral nervous systems. However, the precise temporal dynamics and dose–response relationships of these interacting mechanisms require further elucidation. Future research should focus on optimizing the stimulation protocols to maximize the synergistic effects between anti-inflammatory actions and plasticity induction while addressing the individual variability in treatment responses across different neuropathic pain conditions. These investigations will be crucial for establishing tDCS as a reliable, mechanism-based therapeutic approach in clinical practice.

## Figures and Tables

**Figure 1 biology-14-00892-f001:**
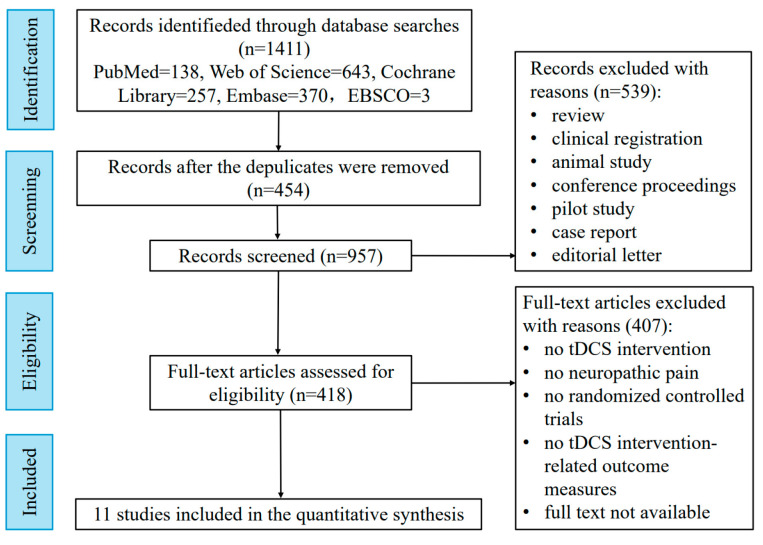
Flowchart of study selection.

**Figure 2 biology-14-00892-f002:**
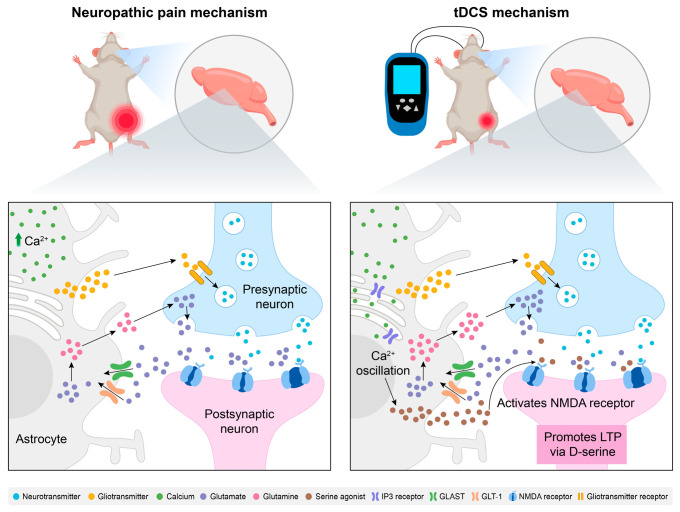
Modulation of neuropathic pain by tDCS: exploring the role of brain inflammatory responses. At the cerebral level, astrocytes upregulate neurotransmitter receptors and release gliotransmitters to potentiate nociceptive signaling. The increased Ca^2+^ levels in astrocytes prompt them to exert their main role by activating specific transporters (GLAST and GLT-1) as glutamate receptors. They also provide glutamine to neurons for the synthesis of glutamate during neural transmission, forming a glutamate–glutamine cycle, thereby enhancing the transmission of pain signals. tDCS enhances the glutamate uptake capacity of astrocytes, inhibits excessive glutamate accumulation, and reduces neuronal excitotoxicity. At the same time, tDCS activates the IP3 signaling pathway in astrocytes, promotes synaptic plasticity dependent on NMDA receptors, inhibits the input of peripheral nociceptive afferent signals, and enhances cortical excitability. Abbreviations: *tDCS*, transcranial direct current stimulation; NMDA, N-methyl-D-aspartate; LTP, long-term potentiation; GLAST, glutamate aspartate transporter; GLT-1, glutamate transporter 1; IP3, inositol 1,4,5-trisphosphate.

**Figure 3 biology-14-00892-f003:**
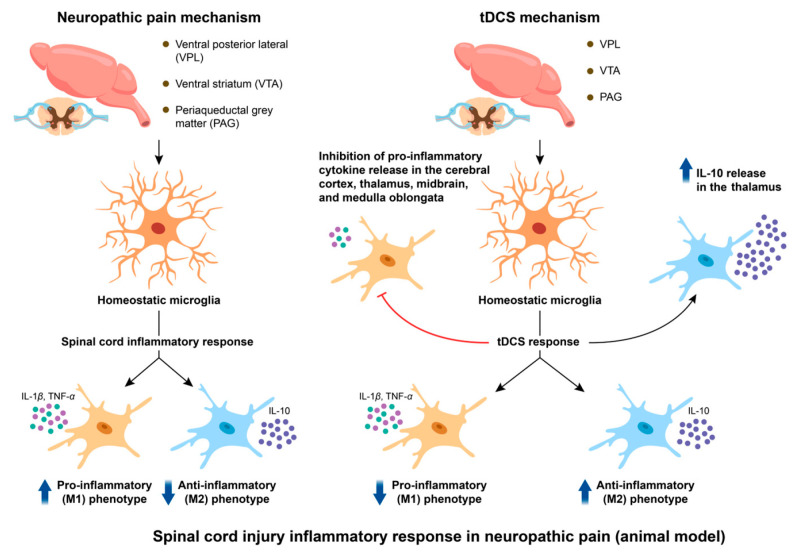
Modulation of neuropathic pain by tDCS: exploring the roles of spinal cord injury and inflammatory responses. After the inflammatory response occurs at the spinal cord level, microglia in the rat brain are activated, and the proportion of M1 phenotype cells increases. The M1 type microglia cells enhance the cytotoxicity of neurons by releasing inflammatory factors such as IL-1β and TNF-a, and promote the transmission of pain signals. tDCS stimulation may promote more microglia to transform from the M1 phenotype to the M2 phenotype. By releasing more anti-inflammatory factors such as IL-10, it enhances the inhibitory effect on pro-inflammatory factors, thereby achieving anti-inflammatory and pain-relieving effects. Abbreviation: tDCS, transcranial direct current stimulation; IL-1β, interleukin-1β; TNF-α, tumor necrosis factor-α; IL-10, interleukin-10.

**Figure 4 biology-14-00892-f004:**
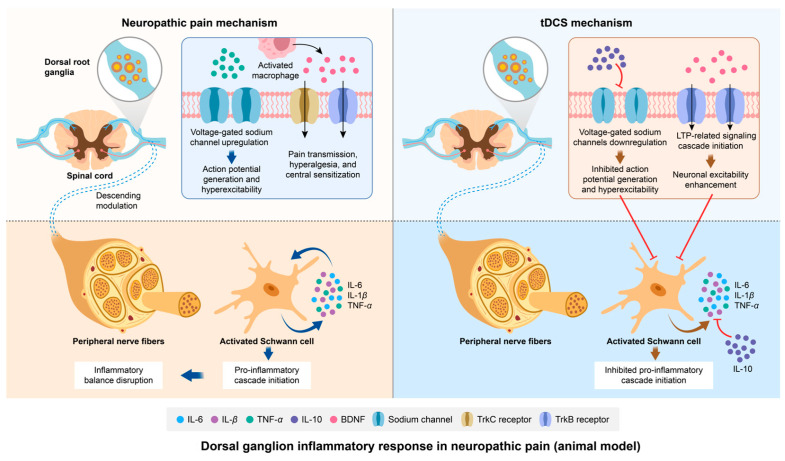
Modulation of neuropathic pain by tDCS: exploring the role of dorsal root ganglion inflammatory responses. After the inflammatory response, TNF-α promotes excessive neuronal excitation by upregulating voltage-gated sodium channels in dorsal root ganglion neurons, leading to neuropathic pain. Additionally, the inflammatory response may activate macrophages to increase the expression of BDNF and TrkC and TrkB receptors, and initiate and maintain mechanical hyperalgesia through the BDNF-TrkB signaling pathway. At the level of peripheral nerves, Schwann cells are activated. They initiate a series of pro-inflammatory cascades and disrupt the balance of inflammation. After tDCS, the downstream inhibitory effect on sodium channels in the DRG is mediated by the upregulation of IL-10 at the spinal cord level. tDCS can induce more BDNF to bind to the high-affinity receptor TrkB, promoting LTP and maintaining normal neuronal excitability. The downstream anti-inflammatory regulatory effect induced by tDCS also occurs at the peripheral nerve level. Abbreviation: *tDCS*, transcranial direct current stimulation; IL-6, interleukin-6; IL-β, interleukin-β; TNF-α, tumor necrosis factor-α; IL-10, interleukin-10, BDNF, brain-derived neurotrophic factor.

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
