# Peer review of "Anti-Inflammatory Pathways Mediating tDCS’s Effects on Neuropathic Pain"

_biology, 2025, doi:10.3390/biology14070892_

Round 1
Reviewer 1 Report
Comments and Suggestions for Authors
This paper provides a comprehensive review of anti-inflammatory mechanisms underlying transcranial direct current stimulation (tDCS) effects in neuropathic pain. While it addresses an important clinical topic, several significant issues limit its quality and impact.
The review lacks details about database searches, inclusion/exclusion criteria, or quality assessment of included studies
Appears to cherry-pick studies supporting the anti-inflammatory hypothesis while potentially overlooking contradictory evidence
Studies are largely presented uncritically without discussing limitations, contradictions, or methodological concerns
Mechanistic Oversimplification: The paper presents overly simplistic causal relationships between tDCS and inflammatory changes without acknowledging the complexity of these interactions. For example, the direct link between cortical tDCS and peripheral nerve inflammation lacks sufficient mechanistic support.
The review mixes high-quality RCTs with case reports and pilot studies without appropriate weighting or quality assessment. Table 1 includes studies with vastly different methodological rigor.
The paper acknowledges mixed clinical outcomes but doesn't adequately address why tDCS shows inconsistent effects if the anti-inflammatory mechanism is as robust as suggested.
Figure 1 is overly simplistic and doesn't accurately represent the complexity of neuroinflammatory networks. The relationships shown appear more definitive than the evidence supports.
Table 2 combines studies with different animal models, stimulation parameters, and outcome measures without proper standardization or comparison.
No meta-analysis or quantitative synthesis is provided despite reviewing multiple studies with similar outcomes.
4. Conceptual Problems
The paper often conflates correlation with causation. Finding altered inflammatory markers after tDCS doesn't necessarily mean these changes mediate the analgesic effects.
Clinical Translation Gap: Limited discussion of why promising preclinical anti-inflammatory findings haven't translated to more consistent clinical outcomes.
- Most cited animal studies use relatively short follow-up periods (7-30 days)
- Stimulation parameters vary widely across studies, making comparisons difficult
- Limited discussion of sham controls and blinding in animal studies
Author Response
We appreciate the reviewer’s suggestions, which will help to increase the scientific quality of the paper and improve the resubmitted version. The corrections according to the comments in the revised manuscript are marked in red.
We will answer in a point-to-point fashion to the comments of the reviewer.
Detailed list of changes made to the manuscript, according to comments from Reviewer#1.
Specific feedback
The review lacks details about database searches, inclusion/exclusion criteria, or quality assessment of included studies
⇒ Thanks for your comments. We adopted a narrative review format to synthesize the historical evolution and milestone breakthroughs in this field, with the aim of identifying future research directions (e.g. Biology 2025, 14(6), 697, Biology 2025, 14(6), 675). Therefore, we did not prioritize formal quality assessment of included studies. We conducted a literature search using the keywords: neuropathic pain, pain, sensory hypersensitivity, mechanisms, tDCS, inflammatory mediators, neuroplasticity, glial, cytokines, and cortex. Relevant studies were ultimately identified through academic databases, including PubMed, Web of Science, Springer, and Ovid. According for your comments, to ensure robustness in evaluating human studies, we exclusively incorporated randomized controlled trials and randomized pilot studies that met our predefined criteria. The principles of the experimental design should include the tDCS stimulation group and the control group. The control groups consist of sham stimulation. The main outcome indicators are related pain scores. Additionally, we have removed some of the lower-quality literature.
Appears to cherry-pick studies supporting the anti-inflammatory hypothesis while potentially overlooking contradictory evidence
⇒ Thanks for your comments. We have added the evidence in this section as follows.
Evidence challenging anti-inflammatory efficacy includes inconsistent IL-10 responses in NP, with levels exhibiting significant fluctuations across etiologies, disease stages, and experimental models. Chronic constriction injury and partial sciatic nerve ligation models demonstrated inverse correlations between IL-10 levels and pain severity, whereas peripheral lesion models showed unaltered IL-10 concentrations in DRG and peripheral nerves despite concurrent inflammation. This suggests that anti-inflammatory interventions solely elevating IL-10 cannot reliably predict therapeutic outcomes.
As noted in the Ellis & Bennett review, while neuroinflammation initiates NP pathogenesis, exclusive anti-inflammatory strategies seem to fail to sustainably suppress NP progression. Transient or ineffective pain relief may occur when pro-inflammatory mediator suppression is implemented without reversing central structural and functional plasticity. Long-term maladaptive neural plasticity is considered likely to be the core determinant of NP chronicity. Consequently, a combined "anti-inflammatory + plasticity-modifying" approach seems to represent a promising therapeutic direction, warranting validation through multimodal assessments integrating neuroimaging and behavioral evaluations. We specifically incorporated the relevant evidence into the “Challenges of tDCS in Neuropathic Pain” section.
Studies are largely presented uncritically without discussing limitations, contradictions, or methodological concerns
⇒ Thanks for your comments. We have phrased this section and specifically incorporated the relevant evidence into the discussion section. We have included the following description regarding the methodological limitations and contradictions.
A fundamental methodological limitation across rodent NP models involves the implementation of inappropriate assessment endpoints. Most cited animal studies employ relatively short observation periods (7-30 days), with pain evaluation exclusively focused on stimulus-evoked hyperreflexia at specific timepoints. This contrasts with the clinical presentation of NP patients, among whom persistent spontaneous pain and sensory deficits predominate. Detection of spontaneous pain behaviors in experimental animals proves particularly challenging. Furthermore, controversy surrounds sham protocols involving rapid current termination after brief stimulation, compounded by unvalidated blinding efficacy due to animals' inability to report perceptual experiences. Uniformity in tDCS modalities, stimulation parameters, and intensity (0.5 mA anodal M1 stimulation) across included animal studies in our article may underlie the validation of the anti-inflammatory pathway hypothesis.
The differences in clinical application may be caused by the following aspects. First of all, the discrepancies in clinical outcomes regarding anti-inflammatory efficacy are attributable to variations in stimulation sites, parameters, and duration within the incorporated clinical investigations. Secondly, heterogeneous responses in NP patients may stem from individual variability, potentially explaining inconsistent tDCS out-comes. Moreover, the small clinical sample sizes further preclude consistent mechanistic validation. For details, please refer to the “Challenges of tDCS in Neuropathic Pain” section of the article.
Mechanistic Oversimplification: The paper presents overly simplistic causal relationships between tDCS and inflammatory changes without acknowledging the complexity of these interactions. For example, the direct link between cortical tDCS and peripheral nerve inflammation lacks sufficient mechanistic support.
⇒ Thanks for your comments. We have revised this section and review more papers to support my claim.
However, a novel theoretical construct termed "dual pro-plasticity" was recently proposed in 2024 research, postulating that inflammation triggered by neural injury induces neuroplastic alterations. These pathological changes subsequently exacerbate glial activation and inflammatory cascades, establishing a self-perpetuating cycle of pathological neuroplasticity and neuroinflammation. At the cerebral level, tDCS-mediated neuroplasticity modulation is evidenced in the hippocampus and motor cortex. One of the mechanism is that the specific transporters of activated astrocytes (excitatory amino acid transporters types 1 and 2 - EAAT1/GLAST and EAAT2/GLT-1) act as the main synaptic glutamate clearance agents,enhanced astrocytic glutamate uptake capacity facilitates glutamine synthesis to reduce excitotoxicity. tDCS also enhances NMDAR-dependent synaptic plasticity via astrocyte-specific Ca²⁺/IP3 pathway activation through serine agonists, with observing suppression of peripheral nociceptive afferent input and enhance the excitability of the cerebral cortex. Spinal cord investigations reveal tDCS inhibited the release of pro-inflammatory factors such as IL-1 and TNFα by M1-type microglia to reduce inflammation and relieve pain, the transformed M2-type microglia release IL-10 to enhance the inhibitory effect of pro-inflammation, achieving the effects of anti-inflammation and pain relief. Within the peripheral nervous system, tDCS-mediated modulation of glial cells at the spinal level induces substantial cytokine release, exerting downstream inhibitory effects on neuroinflammatory responses in injured peripheral nerves. Within dorsal root ganglia which is the peripheral-central interface, the therapeutic effect is attributed to the downstream inhibitory effect on sodium channels mediated by the upregulation of IL-10 at the spinal cord level. Peripheral pain modulation is predominantly mediated through spinal-level downstream regulation, where IL-10 upregulation suppresses pro-inflammatory fac-tors to attenuate nociceptive signaling, thereby reducing immune cell recruitment and activation at injury sites and within the central nervous system. We have included the specific supplementary details in the third section (Inflammatory mechanisms of tDCS for NP). Please refer to the article for more information.
The review mixes high-quality RCTs with case reports and pilot studies without appropriate weighting or quality assessment. Table 1 includes studies with vastly different methodological rigor.
⇒ Thanks for your comments. We adopted a narrative review format to synthesize the historical evolution and milestone breakthroughs in this field, with the aim of identifying future research directions (e.g. Biology 2025, 14(6), 697, Biology 2025, 14(6), 675). According for your comments, to ensure robustness in evaluating human studies, we exclusively incorporated randomized controlled trials and randomized pilot studies that met our predefined criteria. We have deleted case reports for the reason of the low quality. In addition, we have added a randomized controlled study in the second section of the text. The added part is as follows.
A randomized controlled trial in patients with painful diabetic peripheral neuropathy (PDPN) revealed significantly greater analgesic effects following DLPFC stimulation compared to sham interventions, though pain reduction was significantly less pronounced than that achieved with M1 stimulation.
The paper acknowledges mixed clinical outcomes but doesn't adequately address why tDCS shows inconsistent effects if the anti-inflammatory mechanism is as robust as suggested.
⇒ Thanks for your comments. We have included an explanation regarding the reasons for the varying clinical effects of tDCS. Uniformity in tDCS modalities, stimulation parameters, and intensity (0.5 mA anodal M1 stimulation) across included animal studies in our article may underlie the validation of the anti-inflammatory pathway hypothesis. The differences in clinical application may be caused by the following aspects. First of all, the discrepancies in clinical outcomes regarding anti-inflammatory efficacy are attributable to variations in stimulation sites, parameters, and duration within the incorporated clinical investigations. Secondly, heterogeneous responses in NP patients may stem from individual variability, potentially explaining inconsistent tDCS outcomes. Moreover, the small clinical sample sizes further preclude consistent mechanistic validation. For more details of the discussion on clinical translation, we have shown in the “Challenges of tDCS in Neuropathic Pain” section.
Figure 1 is overly simplistic and doesn't accurately represent the complexity of neuroinflammatory networks. The relationships shown appear more definitive than the evidence supports.
⇒ Thanks for your comments. Following reconsideration, Figure 1 was removed and replaced with three newly developed figures depicting the neurophysiological mechanisms of tDCS in neuropathic pain (NP) management. These figures comprehensively characterize NP pathogenesis and tDCS therapeutic mechanisms across three anatomical domains: cerebral systems, spinal cord pathways, and peripheral components—including dorsal root ganglia and peripheral nerves. The relationships among neuroinflammatory networks have been elaborately depicted in the figures.
Table 2 combines studies with different animal models, stimulation parameters, and outcome measures without proper standardization or comparison.
⇒ Thanks for your comments. We standardized and compared the animal experiments included in the research, and then deleted the irrelevant literature. The table has been updated with consensus reached on animal models and stimulation parameters at present. A descriptive synthesis and critical discussion of animal study findings were concurrently incorporated in the discussion which shows as follows. Collectively, current animal studies on tDCS for NP predominantly utilize sciatic nerve injury models. 0.5 mA Bimodal tDCS applied to the murine M1 was observed to suppress neuroinflammatory responses through glial cell activation-mediated anti-inflammatory factor re-lease or enhanced cortical neuronal connectivity elevating neuronal excitability. Alternatively, S1 stimulation with 0.1 mA tDCS attenuated nociceptive signaling via glia-mediated synaptic remodeling. Nevertheless, research on somatosensory cortical tDCS remains nascent, necessitating further mechanistic exploration.
No meta-analysis or quantitative synthesis is provided despite reviewing multiple studies with similar outcomes.
⇒ Thanks for your comments. We adopted a narrative review format to synthesize the historical evolution and milestone breakthroughs in this field, with the aim of identifying future research directions (e.g. Biology 2025, 14(6), 697, Biology 2025, 14(6), 675). Unlike a meta-analysis or systematic review, this approach does not prioritize formal quality assessment of included studies. Therefore, we did not have meta-analysis or similar outcomes.
The paper often conflates correlation with causation. Finding altered inflammatory markers after tDCS doesn't necessarily mean these changes mediate the analgesic effects.
⇒ Thanks for your comments. We have revised in this section and review more papers to support my claim.
At the cerebral level, tDCS-mediated neuroplasticity modulation is evidenced in the hippocampus and motor cortex. One of the mechanism is that the specific transporters of activated astrocytes (excitatory amino acid transporters types 1 and 2 - EAAT1/GLAST and EAAT2/GLT-1) act as the main synaptic glutamate clearance agents,enhanced astrocytic glutamate uptake capacity facilitates glutamine synthesis to reduce excitotoxicity. tDCS also enhances NMDAR-dependent synaptic plasticity via astrocyte-specific Ca²⁺/IP3 pathway activation through serine agonists, with observing suppression of peripheral nociceptive afferent input and enhance the excitability of the cerebral cortex. Spinal cord investigations reveal tDCS inhibited the release of pro-inflammatory factors such as IL-1 and TNFα by M1-type microglia to reduce inflammation and relieve pain, the transformed M2-type microglia release IL-10 to enhance the inhibitory effect of pro-inflammation, achieving the effects of anti-inflammation and pain relief. Within the peripheral nervous system, tDCS-mediated modulation of glial cells at the spinal level induces substantial cytokine release, exerting downstream inhibitory effects on neuroinflammatory responses in injured peripheral nerves. Within dorsal root ganglia which is the peripheral-central interface, the therapeutic effect is attributed to the downstream inhibitory effect on sodium channels mediated by the upregulation of IL-10 at the spinal cord level. Peripheral pain modulation is predominantly mediated through spinal-level downstream regulation, where IL-10 upregulation suppresses pro-inflammatory fac-tors to attenuate nociceptive signaling, thereby reducing immune cell recruitment and activation at injury sites and within the central nervous system. We have included the specific supplementary details in the third section. Please refer to the article for more information.
Clinical Translation Gap: Limited discussion of why promising preclinical anti-inflammatory findings haven't translated to more consistent clinical outcomes.
- Most cited animal studies use relatively short follow-up periods (7-30 days)
- Stimulation parameters vary widely across studies, making comparisons difficult
- Limited discussion of sham controls and blinding in animal studies
⇒ Thanks for your comments. We have separately included relevant explanations for the clinical application part in the “Challenges of tDCS in Neuropathic Pain” section. The “Challenges of tDCS in Neuropathic Pain” section shows as followed.
A fundamental methodological limitation across rodent NP models involves the implementation of inappropriate assessment endpoints. Most cited animal studies employ relatively short observation periods (7-30 days), with pain evaluation exclusively focused on stimulus-evoked hyperreflexia at specific timepoints. This contrasts with the clinical presentation of NP patients, among whom persistent spontaneous pain and sensory deficits predominate. Detection of spontaneous pain behaviors in experimental animals proves particularly challenging. Furthermore, controversy surrounds sham protocols involving rapid current termination after brief stimulation, compounded by unvalidated blinding efficacy due to animals' inability to report perceptual experiences. Uniformity in tDCS modalities, stimulation parameters, and intensity (0.5 mA anodal M1 stimulation) across included animal studies in our article may underlie the validation of the anti-inflammatory pathway hypothesis.
The differences in clinical application may be caused by the following aspects. First of all, the discrepancies in clinical outcomes regarding anti-inflammatory efficacy are attributable to variations in stimulation sites, parameters, and duration within the incorporated clinical investigations. Secondly, heterogeneous responses in NP patients may stem from individual variability, potentially explaining inconsistent tDCS outcomes. Moreover, the small clinical sample sizes further preclude consistent mechanistic validation.

Reviewer 2 Report
Comments and Suggestions for Authors
The scientific paper "Anti-inflammatory Pathways Mediating tDCS Effects in Neuropathic Pain" aimed to summarise the mechanism by which Transcranial direct current stimulation (tDCS) improves Neuropathic pain (NP) by regulating inflammatory mediators. After detailed reading, I can make the following considerations:
1) I recommend increasing the abstract to 200-250 words, incorporating research findings and conclusions.
2) I recommend increasing the number of keywords (up to 10), which favors traceability in literature searches in databases.
3) References in the text must be adjusted. They are not overwritten.
4) Adjust the alignment of the paragraphs. It should be "justified"
5) The images created are important, but their visibility should be improved. They are in low definition and it is difficult to see the texts contained in them.
6) References are not in the correct journal format.
7) Create a critical opinion section before the conclusions. In this section, the authors should contextualize the previous sections, which were too expository and lacked discussion, with their opinions.
8) The tables are adequate.
Author Response
We appreciate the reviewer’s suggestions, which will help to increase the scientific quality of the paper and improve the resubmitted version. The corrections according to the comments in the revised manuscript are marked in red.
We will answer in a point-to-point fashion to the comments of the reviewer.
Detailed list of changes made to the manuscript, according to comments from Reviewer#2.
I recommend increasing the abstract to 200-250 words, incorporating research findings and conclusions.
⇒ Thanks for your comments. We have phrased in this section. The revised summary is as follows.
Neuropathic pain (NP) is a prevalent clinical condition resulting from diseases or injuries affecting the somatosensory system. Conventional analgesics often exhibit limited efficacy, leading to suboptimal therapeutic outcomes. The pathogenesis of NP is com-plex and involves multiple mechanisms. Existing evidence suggests that maladaptive neuronal plasticity plays a central role in NP development. Additionally, emerging re-search highlights the contribution of neuroinflammatory responses mediated by glial cells in the onset of NP and associated sensory hypersensitivity. Among non-invasive neuromodulation techniques, transcranial direct current stimulation (tDCS) has gained prominence as a potential treatment for NP. Numerous studies have demonstrated its analgesic effects; however, the precise regulatory mechanisms remain unclear. Current evidence indicates that tDCS may alleviate NP by enhancing glial-neuronal interactions, which suppress nociceptive signaling pathways and reduce pain sensitivity. The therapeutic effects of tDCS could be mediated through neurotransmitter release targeting specific inflammatory mediators to remodel neuronal plasticity or via anti-inflammatory processes with potential neuroprotective benefits. Recent studies further indicate that tDCS may mitigate NP symptoms by modulating inflammatory mediators. This review summarizes current evidence on the role of tDCS in regulating inflammatory mediators to ameliorate NP.
I recommend increasing the number of keywords (up to 10), which favors traceability in literature searches in databases.
⇒ Thanks for your comments. We have rewritten. The revised keywords are as follows: neuropathic pain; tDCS; inflammatory mediators; neuroplasticity; glial; cytokines; cortex ; spinal cord; dorsal ganglion root; peripheral nerve
References in the text must be adjusted. They are not overwritten.
⇒ Thanks for your comments. We have revisited in the whole paper and revised them.
Minor comments:
Adjust the alignment of the paragraphs. It should be "justified"
⇒ Thanks for your comments. We have revisited the whole paper and revised them.
The images created are important, but their visibility should be improved. They are in low definition and it is difficult to see the texts contained in them.
⇒ Thanks for your comments. We have revised in the paper and improved them.
References are not in the correct journal format.
⇒ Thanks for your comments. We have revised it according to the reference format of our journal.
Create a critical opinion section before the conclusions. In this section, the authors should contextualize the previous sections, which were too expository and lacked discussion, with their opinions.
⇒ Thanks for your comments. We have separately included relevant explanations for the previous sections as our critical opinion in the “Challenges of tDCS in Neuropathic Pain” section. The “Challenges of tDCS in Neuropathic Pain” section shows as followed.
Collectively, current animal studies on tDCS for NP predominantly utilize sciatic nerve injury models. 0.5 mA Bimodal tDCS applied to the murine M1 was observed to suppress neuroinflammatory responses through glial cell activation-mediated anti-inflammatory factor re-lease or enhanced cortical neuronal connectivity elevating neuronal excitability. Alternatively, S1 stimulation with 0.1 mA tDCS attenuated nociceptive signaling via glia-mediated synaptic remodeling. Nevertheless, research on somatosensory cortical tDCS remains nascent, necessitating further mechanistic exploration.
Evidence challenging anti-inflammatory efficacy includes inconsistent IL-10 responses in NP, with levels exhibiting significant fluctuations across etiologies, disease stages, and experimental models. Chronic constriction injury and partial sciatic nerve ligation models demonstrated inverse correlations between IL-10 levels and pain severity, whereas peripheral lesion models showed unaltered IL-10 concentrations in DRG and peripheral nerves despite concurrent inflammation. This suggests that anti-inflammatory interventions solely elevating IL-10 cannot reliably predict therapeutic outcomes.
As noted in the Ellis & Bennett review, while neuroinflammation initiates NP pathogenesis, exclusive anti-inflammatory strategies fail to sustainably suppress NP progression. Transient or ineffective pain relief may occur when pro-inflammatory mediator suppression is implemented without reversing central structural and functional plasticity. Long-term maladaptive neural plasticity is implicated as the core determinant of NP chronicity. Consequently, a combined "anti-inflammatory + plasticity-modifying" approach represents a promising therapeutic direction, warranting validation through multimodal assessments integrating neuroimaging and behavioral evaluations. We specifically incorporated the relevant evidence into the discussion section.
Moreover, studies also found that anodal tDCS reduced multiple neuroinflammatory cytokines and enhanced LTP without concomitant improvements in behavioral pain thresholds or detectable changes in astrocytic activation, suggesting limited glial involvement in synaptic plasticity or anti-inflammatory process. A fundamental methodological limitation across rodent NP models involves the implementation of inappropriate assessment endpoints. Most cited animal studies employ relatively short observation periods (7-30 days), with pain evaluation exclusively focused on stimulus-evoked hyperreflexia at specific timepoints. This contrasts with the clinical presentation of NP patients, among whom persistent spontaneous pain and sensory deficits predominate. Detection of spontaneous pain behaviors in experimental animals proves particularly challenging. Furthermore, controversy surrounds sham protocols involving rapid current termination after brief stimulation, compounded by unvalidated blinding efficacy due to animals' inability to report perceptual experiences. Uniformity in tDCS modalities, stimulation parameters, and in-tensity (0.5 mA anodal M1 stimulation) across included animal studies in our article may underlie the validation of the anti-inflammatory pathway hypothesis.
The differences in clinical application may be caused by the following aspects. First of all, the discrepancies in clinical outcomes regarding anti-inflammatory efficacy are attributable to variations in stimulation sites, parameters, and duration within the incorporated clinical investigations. Secondly, heterogeneous responses in NP patients may stem from individual variability, potentially explaining inconsistent tDCS out-comes. Moreover, the small clinical sample sizes further preclude consistent mechanistic validation. Future high-quality studies integrating multimodal neurophysiological and imaging assessments are imperative to optimize therapeutic protocols, evaluate long-term efficacy, and elucidate underlying mechanisms.
The tables are adequate.
⇒ Thanks for your comments. We have also further enhanced the presentation format of the table.

Round 2
Reviewer 1 Report
Comments and Suggestions for Authors
There are still major concerns:
No systematic review methodology despite attempting comprehensive coverage
Lack of quality assessment for included studies
Poor integration of heterogeneous evidence
Overstated mechanistic claims not supported by current evidence
Inadequate critical analysis of conflicting results
Poor handling of translational limitations between animal and human studies
Persistent redundancy and organizational issues
Tables that mix incomparable studies without proper analysis
Figures that don't effectively communicate the proposed mechanisms
Author Response
We appreciate the reviewer’s suggestions, which will help to increase the scientific quality of the paper and improve the resubmitted version. The corrections according to the comments in the revised manuscript are marked in red.
We will answer in a point-to-point fashion to the comments of the reviewer.
Detailed list of changes made to the manuscript, according to comments from Reviewer#1.
Specific feedback
No systematic review methodology despite attempting comprehensive coverage
⇒ Thanks for your comments. We have included the methodological basis for the literature screening in the article, including the search strategy for the literature, the sources of the literature databases, as well as the inclusion and exclusion criteria. In addition, we have added a flowchart for literature search. For more details on the studies screening process, please refer to “Therapeutic Effects of tDCS on Neuropathic Pain Management” section of the context.
Relevant studies were ultimately identified through academic databases included PubMed, Web of Science, Cochrane Library, Embase and EBSCO, performed in March 2025, with no restriction of publication year and language. The search strategy used a mix of Mesh (or Publication Type) and free text terms from the following 2 group. 1. " tDCS [Mesh]”, “Anodal Stimulation Transcranial Direct Current Stimulation”, “An-odal Stimulation tDCS*”, “Cathodal Stimulation Transcranial Direct Current Stimulation”, “Cathodal Stimulation tDCS*”, “Transcranial Alternating Current Stimulation”, “Transcranial Random Noise Stimulation”, “Repetitive Transcranial Electrical Stimulation”, “Transcranial Electrical Stimulation*”. 2. “Neuralgias [Mesh]”, “Neurodynia*”, “Neuropathic Pain*”, “Nerve Pain*”, “Paroxysmal Nerve Pain*”, “Atypical Neuralgia*”, “Iliohypogastric Nerve Neuralgia*”, “Ilioinguinal Neuralgia *”, “Perineal Neuralgia*”, “Stump Neuralgia*”, “Supraorbital Neuralgia*”, “Vidian Neuralgia*”. To ensure robustness in evaluating human studies, we have respectively set the inclusion and exclusion criteria as follows. The inclusion criteria include: (1) patients diagnosed with neuropathic pain (no restriction on sex and age); (2) The interventions were tDCS intervention; (3) The comparators were sham stimulation. (4) The outcome measure was changes in pain intensity. (5) The study design was RCT. The exclusion criteria include: (1) duplicate items; (2) if articles were review, clinical registration, animal study, conference, pilot study, case report, editorial letters or full text not available. Figure 1 presents the flow chart of the search for bibliographic references of this review.
Lack of quality assessment for included studies
⇒ Thanks for your comments. We incorporated a literature search strategy into the article, particularly adding inclusion and exclusion criteria to assess the quality of our current study. After considering, we have excluded all pilot studies and and have included three new studies (Fregni et al., Ayache et al., Gonzá-lez-Zamorano et al.) in the tables. The added content is as follows, and the detailed information can be found in the “Therapeutic Effects of tDCS on Neuropathic Pain Management” section of the article.
Relevant studies were ultimately identified through academic databases included PubMed, Web of Science, Cochrane Library, Embase and EBSCO, performed in March 2025, with no restriction of publication year and language. The search strategy used a mix of Mesh (or Publication Type) and free text terms from the following 2 group. 1. " tDCS [Mesh]”, “Anodal Stimulation Transcranial Direct Current Stimulation”, “An-odal Stimulation tDCS*”, “Cathodal Stimulation Transcranial Direct Current Stimulation”, “Cathodal Stimulation tDCS*”, “Transcranial Alternating Current Stimulation”, “Transcranial Random Noise Stimulation”, “Repetitive Transcranial Electrical Stimulation”, “Transcranial Electrical Stimulation*”. 2. “Neuralgias [Mesh]”, “Neurodynia*”, “Neuropathic Pain*”, “Nerve Pain*”, “Paroxysmal Nerve Pain*”, “Atypical Neuralgia*”, “Iliohypogastric Nerve Neuralgia*”, “Ilioinguinal Neuralgia *”, “Perineal Neuralgia*”, “Stump Neuralgia*”, “Supraorbital Neuralgia*”, “Vidian Neuralgia*”. To ensure robustness in evaluating human studies, we have respectively set the inclusion and exclusion criteria as follows. The inclusion criteria include: (1) patients diagnosed with neuropathic pain (no restriction on sex and age); (2) The interventions were tDCS intervention; (3) The comparators were sham stimulation. (4) The outcome measure was changes in pain intensity. (5) The study design was RCT. The exclusion criteria include: (1) duplicate items; (2) if articles were review, clinical registration, animal study, conference, pilot study, case report, editorial letters or full text not available. Figure 1 presents the flow chart of the search for bibliographic references of this review.
Poor integration of heterogeneous evidence
⇒ Thanks for your comments. Firstly, we provided separate explanations for the varying efficacy of tDCS in different types of NPs and different brain regions as follows. For detailed information, please refer to “Therapeutic Effects of tDCS on Neuropathic Pain Management” section of the article. Secondly, we have conducted a detailed discussion on the heterogeneity issue existing between clinical and animal studies, and the additional contents are as follows. For detailed information, please refer to “Translational Limitations and Future Directions” section of the article.
The differential efficacy of tDCS across various subtypes of NP is likely attributable to fundamental distinctions in their underlying pathophysiological mechanisms. Phantom limb pain is predominantly associated with central neural reorganization, characterized by pronounced cortical plasticity abnormalities, rendering stimulation of the M1 more responsive in eliciting analgesic effects. In contrast, peripheral NP involves distinct neural circuits, wherein modulation via M1-targeted tDCS may exert a comparatively limited therapeutic impact. Variability in stimulation intensity, as well as inter-individual differences in cortical excitability, have also been implicated as potential confounding factors influencing treatment outcomes. Validation of these observations requires further investigation through tDCS interventions tailored to specific NP subtypes, with additional focus on optimizing stimulation parameters such as intensity, frequency, and duration.
The variability in analgesic efficacy observed across different stimulation targets is hypothesized to result from the functional specialization of these brain regions in pain processing. The M1 area is integrally connected with pain-perception networks, including the thalamus, brainstem, and spinal cord, suggesting a more direct influence on nociceptive signal modulation and, consequently, a more pronounced analgesic response. In comparison, the DLPFC is primarily implicated in the affective and cognitive dimensions of pain, indicating a greater potential for alleviating anxiety and discomfort, but a relatively limited capacity for direct pain suppression. There is a pressing need for large-scale, multicenter, randomized controlled trials to systematically evaluate the therapeutic efficacy of various cortical targets in the treatment of neuropathic pain.
The translation of tDCS effects from animal models to clinical applications for neuropathic pain faces several significant challenges. Animal studies predominantly utilize standardized nerve injury models that induce mechanical allodynia, while human neuropathic pain encompasses diverse etiologies with complex psychosocial dimensions that are difficult to replicate in rodents. This fundamental difference in pain phenotypes limits the generalizability of preclinical findings to the heterogeneous clinical population. Species-specific variations in neuroimmune responses present another critical barrier. While rodent studies demonstrate consistent patterns of glial activation and cytokine modulation following tDCS, human neuroinflammatory pathways may respond differently due to evolutionary divergence in immune system regulation. The blood-brain barrier's differential permeability across species further complicates direct translation of anti-inflammatory effects observed in animal models. Methodological disparities between preclinical and clinical studies contribute to translational challenges. Animal research typically employs prolonged, low-intensity stimulation protocols that differ substantially from clinical tDCS parameters in both dose and application. The lack of standardized protocols for current density adjustment across species makes it difficult to determine equivalent effective doses between rodents and humans. Outcome measurement discrepancies represent an additional limitation. Preclinical studies rely primarily on evoked pain behaviors, whereas clinical trials must incorporate subjective pain reports and functional assessments that reflect the multidimensional nature of human pain experience. This measurement gap underscores the need for bridging biomarkers that can objectively track treatment effects across species, such as neuroimaging markers of cortical excitability or serum cytokine profiles.
Overstated mechanistic claims not supported by current evidence
⇒ Thanks for your comments. We have added the tDCS-based explanation of the NP treatment mechanism based on the "dual pro-plasticity" theory. The "dual pro-plasticity" theory is a relatively comprehensive theoretical model which provides a general overview of the etiological mechanism of NP. The revised content is as follows. Detailed modifications can be found in the abstract and the conclusion section of the article.
The revised abstract section is as follows: Neuropathic pain (NP) is a prevalent clinical condition resulting from diseases or injuries affecting the somatosensory system. Conventional analgesics often exhibit limited efficacy, leading to suboptimal therapeutic outcomes. The pathogenesis of NP is complex and involves multiple mechanisms. Existing evidence suggests that maladaptive neuronal plasticity plays a central role in NP development. Additionally, emerging research highlights the contribution of neuroinflammatory responses mediated by glial cells in the onset of NP and associated sensory hypersensitivity. Among non-invasive neuromodulation techniques, transcranial direct current stimulation (tDCS) has gained prominence as a potential treatment for NP. Numerous studies have demonstrated its analgesic effects; however, the precise regulatory mechanisms remain unclear. Current evidence indicates that tDCS may alleviate NP by enhancing glial-neuronal interactions, which suppress nociceptive signaling pathways and reduce pain sensitivity. The reciprocal modulation between tDCS-mediated anti-inflammatory actions, evidenced by decreased pro-inflammatory cytokines and increased anti-inflammatory mediators, and its facilitation of adaptive neural plasticity represents a particularly compelling therapeutic axis. This review elucidates inflammatory regulation by tDCS as a fundamental mechanism for NP alleviation, while delineating important unresolved questions regarding these complex interactions.
The revised conclusion section is as follows: tDCS demonstrates multimodal therapeutic potential for NP by concurrently targeting neuroinflammation and synaptic plasticity, with bidirectional crosstalk between these two regulatory mechanisms. The intervention appears to simultaneously suppress pro-inflammatory cytokines (IL-1β, TNF-α) while enhancing anti-inflammatory mediators (IL-10), creating a favorable microenvironment for neural reorganization. This immunomodulation interacts bidirectionally with tDCS-induced neuroplastic changes, including NMDA receptor-dependent synaptic remodeling mediated through astrocytic calcium/IP3 signaling pathways. The concurrent regulation of ion channel homeostasis in dorsal root ganglia suggests a comprehensive "top-down" mechanism spanning central and peripheral nervous systems. However, the precise temporal dynamics and dose-response relationships of these interacting mechanisms require further elucidation. Future research should focus on optimizing stimulation protocols to maximize the synergistic effects between anti-inflammatory action and plasticity induction, while addressing individual variability in treatment response across different neuropathic pain conditions. These investigations will be crucial for establishing tDCS as a reliable, mechanism-based therapeutic approach in clinical practice.
Inadequate critical analysis of conflicting results
⇒ Thanks for your comments. We have revised the article again. The revised content is as follows. For specific details, please refer to the “Therapeutic Effects of tDCS on Neuropathic Pain Management” section of the article.
The differential efficacy of tDCS across various subtypes of NP is likely attributable to fundamental distinctions in their underlying pathophysiological mechanisms. Phantom limb pain is predominantly associated with central neural reorganization, characterized by pronounced cortical plasticity abnormalities, rendering stimulation of the M1 more responsive in eliciting analgesic effects. In contrast, peripheral NP involves distinct neural circuits, wherein modulation via M1-targeted tDCS may exert a comparatively limited therapeutic impact. Variability in stimulation intensity, as well as inter-individual differences in cortical excitability, have also been implicated as potential confounding factors influencing treatment outcomes. Validation of these observations requires further investigation through tDCS interventions tailored to specific NP subtypes, with additional focus on optimizing stimulation parameters such as intensity, frequency, and duration.
The variability in analgesic efficacy observed across different stimulation targets is hypothesized to result from the functional specialization of these brain regions in pain processing. The M1 area is integrally connected with pain-perception networks, including the thalamus, brainstem, and spinal cord, suggesting a more direct influence on nociceptive signal modulation and, consequently, a more pronounced analgesic response. In comparison, the DLPFC is primarily implicated in the affective and cognitive dimensions of pain, indicating a greater potential for alleviating anxiety and discomfort, but a relatively limited capacity for direct pain suppression. There is a pressing need for large-scale, multicenter, randomized controlled trials to systematically evaluate the therapeutic efficacy of various cortical targets in the treatment of neuropathic pain.
While combination therapies integrating tDCS with behavioral interventions show promising initial results, several fundamental limitations currently constrain the interpretation of these findings. First, the majority of studies fail to implement adequate sham controls for both the neuromodulation and behavioral components, making it difficult to isolate specific treatment effects from placebo responses. Second, the substantial variability in reported effect sizes (ranging from 30-70% pain reduction) suggests potential confounding by unmeasured variables such as treatment adherence, pain chronicity, or individual differences in neuroanatomy. Most critically, the putative neural mechanisms underlying these synergistic effects remain largely hypothetical. For example, this mechanistic uncertainty is well illustrated by Kumru et al.'s visual illusion study, where the observed analgesic effects likely resulted from attentional modulation through frontoparietal circuitry rather than the intended sensorimotor network targeting. Such a fundamental discrepancy in mechanism engagement has profound implications for developing precisely targeted combination therapies.
Poor handling of translational limitations between animal and human studies
⇒ Thanks for your comments. We have added the evidence in the “Translational Limitations and Future Directions” section as follows.
The translation of tDCS effects from animal models to clinical applications for neuropathic pain faces several significant challenges. Animal studies predominantly utilize standardized nerve injury models that induce mechanical allodynia, while human neuropathic pain encompasses diverse etiologies with complex psychosocial dimensions that are difficult to replicate in rodents. This fundamental difference in pain phenotypes limits the generalizability of preclinical findings to the heterogeneous clinical population. Species-specific variations in neuroimmune responses present another critical barrier. While rodent studies demonstrate consistent patterns of glial activation and cytokine modulation following tDCS, human neuroinflammatory pathways may respond differently due to evolutionary divergence in immune system regulation. The blood-brain barrier's differential permeability across species further complicates direct translation of anti-inflammatory effects observed in animal models. Methodological disparities between preclinical and clinical studies contribute to translational challenges. Animal research typically employs prolonged, low-intensity stimulation protocols that differ substantially from clinical tDCS parameters in both dose and application. The lack of standardized protocols for current density adjustment across species makes it difficult to determine equivalent effective doses between rodents and humans. Outcome measurement discrepancies represent an additional limitation. Preclinical studies rely primarily on evoked pain behaviors, whereas clinical trials must incorporate subjective pain reports and functional assessments that reflect the multidimensional nature of human pain experience. This measurement gap underscores the need for bridging biomarkers that can objectively track treatment effects across species, such as neuroimaging markers of cortical excitability or serum cytokine profiles.
Persistent redundancy and organizational issues
⇒ Thanks for your comments. We have revised the context for the organizational issues and have already made deletions and revisions to all the redundant parts.
Tables that mix incomparable studies without proper analysis
⇒ Thanks for your comments. We have excluded all the pilot studies and have included three new studies (Fregni et al., Ayache et al., Gonzá-lez-Zamorano et al.) in the tables based on the added quality assessment criteria. The detailed content can be found in Table 1.
Figures that don't effectively communicate the proposed mechanisms
⇒ Thanks for your comments. We have added textual explanations for pathways below each figure as follows.
Figure 2. Modulation of Neuropathic Pain by tDCS: Exploring the Role of Brain Inflammatory Responses. At the cerebral level, astrocytes upregulated neurotransmitter receptors and releasing gliotransmitters to potentiate nociceptive signaling. The increased Ca²⁺ levels in astrocytes prompts them to exert their main role by activating specific transporters (GLAST and GLT-1) as glutamate receptors. They also provide glutamine to neurons for the synthesis of glutamate during neural transmission, forming a glutamate-glutamine cycle, thereby enhancing the transmission of pain signals. tDCS enhances the glutamate uptake capacity of astrocytes, inhibits excessive glutamate accumulation, and reduces neuronal excitotoxicity. At the same time, tDCS activates the IP3 signaling pathway in astrocytes, promotes synaptic plasticity dependent on NMDA receptors, inhibits the input of peripheral nociceptive afferent signals, and enhances cortical excitability. Abbreviation: tDCS, transcranial direct current stimulation; NMDA, N-methyl-D-aspartate; LTP, long-term potentiation; GLAST, glutamate aspartate transporter; GLT-1, glutamate transporter 1; IP3, inositol 1,4,5-trisphosphate.
Figure 3. Modulation of Neuropathic Pain by tDCS: Exploring the Role of Spinal Cord Injury Inflammatory Responses. After the inflammatory response occurs at the spinal cord level, microglia in the rat brain are activated, and the proportion of M1 phenotype cells. The M1 type microglia cells enhance the cytotoxicity of neurons by releasing inflammatory factors such as IL-1β and TNF-α, and promote the transmission of pain signals. tDCS stimulation may promote more microglia to transform from the M1 phenotype to the M2 phenotype. By releasing more anti-inflammatory factors such as IL-10, it enhances the inhibitory effect on pro-inflammatory factors, thereby achieving anti-inflammatory and pain-relieving effects. Abbreviation: tDCS, transcranial direct current stimulation; IL-1β, interleukin-1β; TNF-α, tumor necrosis factor-α; IL-10, interleukin-10.
Figure 4. Modulation of Neuropathic Pain by tDCS: Exploring the Role of Dorsal Ganglion Inflammatory Responses. After the inflammatory response, TNF-α promotes excessive neuronal excitation by upregulating voltage-gated sodium channels in dorsal root ganglion neurons, leading to neuropathic pain. Additionally, the inflammatory response may activate macrophages to increase the expression of BDNF in TrkC and TrkB receptors, and initiate and maintain mechanical hyperalgesia through the BDNF-TrkB signaling pathway. At the level of peripheral nerves, Schwann cells are activated. They initiated a series of pro-inflammatory cascades and disrupted the balance of inflammation. After tDCS, the downstream inhibitory effect on sodium channels in DRG is mediated by the upregulation of IL-10 at the spinal cord level. tDCS can induce more BDNF to bind to the high-affinity receptor TrkB, promoting LTP and maintaining the normal neuronal excitability. The downstream anti-inflammatory regulatory effect induced by tDCS also occurs at the peripheral nerve level. Abbreviation: tDCS, transcranial direct current stimulation; IL-6, interleukin-6; IL-β, interleukin-β; TNF-α, tumor necrosis factor-α; IL-10, inter-leukin-10, BDNF, brain-derived neurotrophic factor.
Reviewer 2 Report
Comments and Suggestions for Authors
Thank you for making the suggested changes. I recommend accepting it for publication.
Author Response
Thank you for accepting our manuscript for publication.
Round 3
Reviewer 1 Report
Comments and Suggestions for Authors
The manuscript improved significantly.